# Implementation of Blended Learning during COVID-19

**Santiago Batista-Toledo** * and **Diana Gavilan**

Department of Marketing, Complutense University of Madrid, 28040 Madrid, Spain
* Correspondence: sabatist@ucm.es

**Definition:** Blended learning (BL) is a teaching model that combines face-to-face activities in the classroom with activities outside the classroom through the introduction of technology that is computer-based, distance, or mobile learning, among others. There are several BL models to adopt, depending on the importance and extent to which the technology is used. It brings great benefits to the learner and involves teachers in the design of new teaching methods.

**Keywords:** blended learning; synchronous learning; asynchronous learning experience; COVID-19; higher education

## 1. Introduction

The COVID-19 pandemic precipitated a profound change in the global educational system, transforming the way in which education was delivered. According to UNESCO [1], 81.8% of students globally, at different academic levels, were affected by the total or partial closure of education centres. Initially, due to the lockdown, the education centers were forced to shut down, thereby shifting teaching to online platforms and resulting in an increase of up to 200% in the use of educational applications [2]. As the health situation improved, students gradually moved back to face-to-face classrooms, giving rise, in many cases, to the teaching model known as blended learning (BL), which combines face-to-face and online learning strategies.

BL allowed a partial return to the desired 'normality,' respecting the current sanitary measures of social distancing and seating capacity. However, BL also posed a challenge to the actors involved in the learning process—teachers, students, and institutions—given the infrastructure and organization required to successfully carry it out.

In universities, the implementation of BL has been prolonged more than in other levels of education, such as primary and secondary since higher education seemed to be less vulnerable to the need to return to the face-to-face teaching model. Information technology has changed the role of faculty members and the teaching–learning process itself [3]; therefore, in higher education institutions, students were prone to combine face-to-face instruction with on-line instructional resources, understanding the physical presence of the professor as a complementary form of communication [4]. This has had a more visible impact, affecting the characteristics linked to the essence of universities, such as student mobility, research, and knowledge transfer [5].

In addition, BL has had an impact on several aspects of university life, such as teaching, learning, social relations, costs, and the use of technology, all of which have turned the educational community into a true laboratory of innovation. This new educational context and the existence of new health crises resulting in situations like the one we have experienced make it necessary to study and understand how the COVID-19 pandemic has affected the educational framework, as well as its academic implications (performance, costs, quality of teaching, etc.). This will make it possible, firstly, to determine the real viability of this teaching model in educational and organizational terms and, consequently, to develop measures and policies that will favor its correct functioning in the future.

In fact, as can be seen in Figure 1, in a survey conducted in 2020 in 29 countries around the world about the future of higher education, a majority of the participants stated that they considered teaching in universities would be carried out under the BL modality five years from now [6]. The survey showed that the respondents considered blended and online teaching as normal and key, which emphasizes the importance of understanding this type of modality.

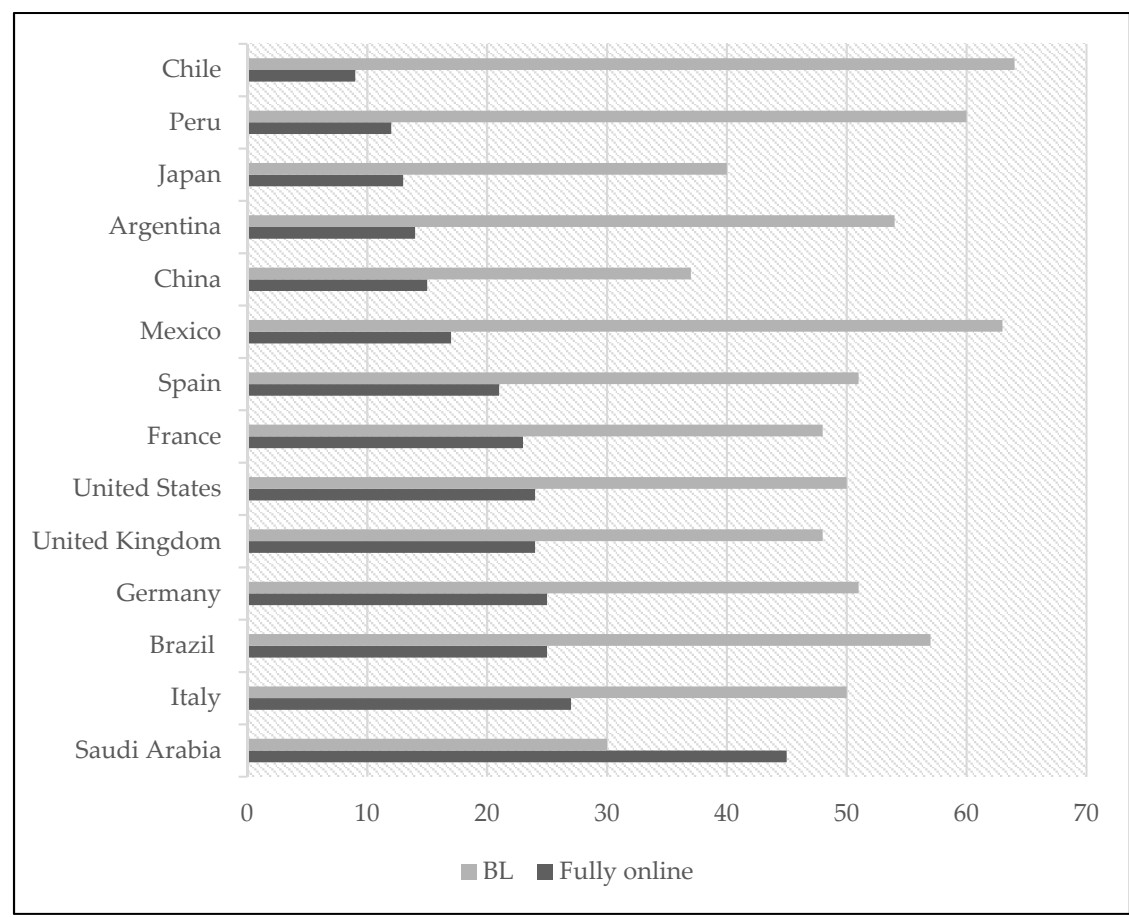

**Figure 1.** Sample of responses from the survey about the teaching modalities that will be used in the future for higher education.

It is also worth mentioning the different measures that were adopted by the countries considered. In general terms, universities extended deadlines for assignments or exams and modified evaluation methods. For instance, in Argentina, an online portal was developed with content for teachers and students, and in Spain, computer resources were purchased for vulnerable families. In the field of research, United Kingdom and United States created platforms to promote collaboration and inter-university research on COVID-19 [5].

The objective of this study is to show the impact of BL implementation in higher education during the pandemic. Although BL has had a gradual implementation, the pandemic forced universities to implement this type of teaching immediately to the whole community of higher education. Therefore, it is of interest to analyze the impact of BL in the experience of students and professors during COVID-19. To this end, the results of numerous academic studies are collected from the perspective of the experience of students, professors, and universities. It concludes with the challenges posed by BL during the pandemic and in the post-pandemic future. In this regard, this chapter conceptualizes the BL model, its dimensions, and the way it may be implemented. Moreover, it describes the consequences of BL experienced by both teachers and students during the COVID-19 pandemic. Finally, the technical and infrastructural implications that universities have

faced while implementing BL are presented, as well as the challenges institutions will face in managing an adequate BL approach that meets their needs. To this end, the research carried out during the pandemic between 2020 and 2022 is analyzed. Understanding what the experience of BL has been like until now is crucial for the development of measures and policies favoring its correct functioning in the future.

## 2. Description of Blended Learning

The beginnings of BL date back to the mid-19th century when Sir Issac Pitman introduced distance learning by sending materials to his students through the mail. However, it was not until the 1960s and 1970s that the first incorporation of technology as a means of distance learning occurred. In particular, Stanford University, which was one of the first organizations to adopt distance learning, used a video network model that allowed its students to learn without the physical presence of a professor. As early as the 1990s, specific learning systems—such as CD-ROMs and web resources—were actually deve-loped, allowing teaching and training outside the classroom, thereby enhancing this type of teaching. Several continuous technological advances, from the early 2000s to the present, have been shaping what we know today as BL [7].

The definition of the phenomenon of BL itself is reflected in its evolution over the years. One of the first academic definitions of BL was from Garrison and Kanuka [8], who defined BL as 'the thoughtful integration of classroom face-to-face learning experiences with online learning experiences' (page 96). Likewise, the authors stated that for BL to take place, there must be a balance between face-to-face and online teaching methods so that neither dominates the other. However, this definition—as well as BL itself—has evolved into broader concepts closer to the definitions given by Goncharov et al. [9] or Siripongdee et al. [10]. Therefore, BL is a process of assimilating knowledge, skills, and methods through a combination of face-to-face, computer-based, distance, and mobile learning, creating an environment that allows for the incorporation and combination of any technology in or outside the classroom. Consequently, a wide variety of BL models have emerged [9]:

- Station rotation: Face-to-face and technology-based activities are alternated while students work in small groups. The teacher decides the content of the activity and gives the students a suitable tool.
- Lab rotation: Students are first taught by a teacher in a classroom and then work individually in a computer class to consolidate and build upon what they have already learned in class.
- Flipped: The student learns the theoretical material independently outside the classroom, deciding on how and when to learn. The classroom focuses on active learning, where the teacher encourages the students to clear their doubts and challenge their understanding.
- Individual rotation: Each student has a specific learning program, rotating through different activities based on their interests and abilities.
- Flexible: Students learn independently outside the classroom using online resources. Once in the classroom, they discuss what they have learned.
- Self-mix: Students choose the type of online learning they want to supplement the face-to-face classes with.
- Enriched virtual: Students mainly complete their learning online but spend several hours face-to-face with a teacher solving doubts and deepening their understanding of the concepts.

These models show the evolution of BL to satisfy the emerging needs of students and teachers. The mixture of technology-based resources with the presence of a teacher varies within each model. Eventually, technology outweighs the teacher, who becomes a guide and support for the student. This gives the student greater autonomy and decision-making ability in the learning process. This evolution also showcases the main features of BL, which are face-to-face and self-paced, as well as distant and ubiquitous [10].

In this regard, the BL case studies during COVID-19 included the use of digital applications to replace blackboards and the classroom and the encouragement of group

work or the leadership assumed by the different members of the university community in carrying out their activities [11]. Other case studies pointed to the improvement in performance compared to the face-to-face modality [12].

### 3. The Impact of Blended Learning in Higher Education

The impact of BL has been a topic of continuous research interest in recent years around the world due to its importance in higher education. In this sense, and for the purpose of this paper, in addition to the research carried out during the pandemic, the research completed prior to the pandemic will be described in order to understand and compare the results of this phenomenon over time. Likewise, it is specified that the COVID-19 period frames the studies carried out from February 2020 to September 2021, corresponding to the phases of implementation of sanitary measures, whereas the pre-COVID-19 period refers to those studies prior to the indicated dates.

#### 3.1. Pre-COVID-19 Period

Academically, student autonomy is one of the main benefits of BL as it allows the students to decide their own learning pace [7]. This increases the student's individual work and extends their learning outside the classroom, thus generating a continuous learning process instead of limiting learning process to the classroom [13]. Moreover, this dissociation of learning within the classroom and the addition of technological resources have made it possible for teachers to communicate with students outside of the classroom [7]. It involves a learner-centered learning style, which enhances student performance and skill achievement [14]. Specifically, compared to conventional lectures, the learner's abilities to process and acquire knowledge and solve problems are emphasized.

However, Ma and Lee [15], in a comparison between the three teaching models—face-to-face, online, and BL—stated that even though BL increases student satisfaction with respect to the other teaching models, students engagement and performance in BL model do not show a significant improvement, contradicting what is mentioned above.

One of the main characteristics of BL is the incorporation of technology. This fact favors the development of students and teachers with greater technological skills [7]. This was demonstrated by Hadiyanto et al. [14] in their research that indicated significant differences in the acquisition of digital skills in BL with respect to face-to-face classes.

For universities, it generated greater flexibility and the capacity for greater reach as it enabled them to offer more courses and programs to a larger volume of students [13]. This, in turn, led to more efficient use of the available resources [9].

From a social point of view, BL occasionally means that students are not physically present in the same place but do more individualized work. However, communication between students and teachers, as well as among students themselves, is increased. This is because BL provides the opportunity, through online activities, for social interactions to take place outside of the classroom, which has a positive impact on the students' social and communicative skills [14]. These interactions, either synchronous or asynchronous, result in the creation of a community between students and teachers [9].

Negatively, BL presents problems in terms of accessibility and equality. The need for computer equipment and internet access is a problem for students with fewer resources. It is for this reason that universities, especially public universities that guarantee equal access to the educational system for all, must support this group of students in order to maintain quality education [16].

#### 3.2. COVID-19 Period

Students were satisfied with BL because they found it to be an environment conducive to learning, as this modality allowed for the incorporation of interesting content in face-to-face classes and effective interaction with the faculty [17]. However, students did not see this incorporation of interesting content in virtual classes [18]. This modality, in comparison with the fully online teaching modality, gave them the opportunity to have at any time and

place the materials explained in class [19], in addition to having the possibility to avail the teacher's support outside of the classroom [20].

Individually, the students highlighted the flexibility they had with BL to organize their time and follow the pace of study that best suited them [17,18,21]. In particular, the fact that they did not have to travel to the university for classes saved them time that they invested in studying or in other activities. This allowed the students to be more liberated and more engaged in their learning, thus improving their academic performance [18,19]. However, they found it difficult to combine online and face-to-face activities as, in many cases, both of these overlapped [17,22].

Despite all of this, in their study, Gazica et al. [23] concluded that students reported higher levels of motivation, commitment, and fulfillment of expectations created in the face-to-face modality and preferred it over BL.

In the case of teachers, there is less research. Even so, teachers consider BL as a desirable future option due to the richness that the combination of face-to-face and online classes would bring to the learning experience [24]. As for students, introducing new content to the classes was one of the main features highlighted by teachers. Performing online activities on what was explained in the class, such as discussion forums or quizzes, allowed teachers to carry out teaching methods that were adapted to the needs of all students [25].

The web resources and online lecture recordings were highlighted by the teachers as a positive of BL, as those allowed the students to view the lecture materials as many times as they wanted. This, according to the teachers, improved the students' understanding of the subject matter [25].

The entry of technology and its importance in the development of the different activities have had implications on students. Many students perceived that BL made it difficult for them to adequately demonstrate their knowledge. This was because the students did not have the necessary training to handle and solve the problems using the different tools, generating in them the assumption that they did not show an adequate performance [21].

The teachers also found it difficult to use the online tools due to their lack of technological knowledge. One of the main problems was caused by having to request the students to keep their cameras active during online lectures, given the implications in terms of the privacy that this entailed [26]. Finally, some researchers have reported teachers' concerns about the threatening role that the entry of technology poses to their status. Specifically, they argue that modalities such as BL, which give students greater autonomy in their learning, may reduce their importance and need as actors in training. Consequently, these teachers question the capacity of online materials as substitutes for face-to-face classes [27].

In terms of socialization, the advent of BL has reduced in-person interactions among students, leading, in some cases, to a sense of social isolation among the students [21]. However, online communications in the realization of lecture activities, together with the use of social networks, have mitigated the feeling of isolation expressed by students [28]. This has strengthened relationships among the students and between the students and the teachers, creating a community that encouraged collaboration [17]. An example of this was the creation of online forums where students shared ideas and which brought teachers closer to students [27], resulting in classroom dynamics favorable for greater interactions [25].

BL has also allowed students to experience a comfortable transition from fully online to face-to-face teaching, making them feel confident about the existing health situation [22]. It was also favorable for students with mobility problems or disabilities as they were no longer obliged to travel to the university to receive the teaching, and this did not affect their learning process [18].

## 4. The Challenges and Opportunities of BL Implementation in Higher Education during the COVID-19 Period and Post-Pandemic

Table 1 presents the main results of both periods, showing the overlap in academic-related results and the differences in the technological and social aspects.

**Table 1.** Summary of the main results for each period analyzed.

|  | **Pre COVID-19 Period** | **COVID-19 Period** |  |
|---|---|---|---|
| Academic experience | Student autonomy | Student autonomy | [7,17,19] |
|  | Greater performance | Greater performance | [14,18,19] |
|  | New teaching methods | New teaching methods | [7,25] |
| Technological experience | Lack of technological knowledge | Lack of technological knowledge | [14,21] |
|  | Acquisition of digital skills | Loss of privacy | [14,26] |
| Social experience | Greater communication | Greater communication | [7,9,28] |
|  | Problems of accessibility and equality | Favorable for students with mobility problems | [16,18] |

Nevertheless, COVID-19 and the arrival of BL compelled universities to face a series of challenges for which they were not prepared. Since the beginning of the health crisis, universities had to ensure continuity in teaching, necessitating training programs for teachers on the different online tools, as well as the new pedagogical ways of teaching [29,30]. Simultaneously, universities had to invest in infrastructure and equipment to provide their centers, students, and teachers with the necessary resources to ensure that the learning process was effectively carried out in the new environment [31,32].

The universities had to develop strategies to ensure that everyone could access and attend their classes while safeguarding the students' safety from possible contagion [33]. Additionally, they had to address the potential digital divide that could affect some of the students through the provision of computer aids and equipment, as well as by developing legal frameworks to regulate issues related to the introduction of technology, such as privacy concerns (due to the obligation to activate cameras) and student assessments [34].

However, the implementation of BL has brought several opportunities for universities and educational institutions. The pandemic emphasized the development of contingency plans to be better prepared for future situations similar to this one and in enacting a rapid response. Blended learning highlighted the need to create continuous training plans for teachers and students using online tools. Likewise, in the same way the methods of teaching are changing, the students' ways of choosing the university where they will develop their studies are also changing, taking into consideration the capacity of the universities to face adverse situations in a manner that does not affect their learning. Therefore, having pre-established plans can become a determining factor in students' future university choices [32].

In this regard, the experience with BL during COVID-19 sheds light on the future of universities. Blended learning could be a potential way of attracting students [31], regardless of their location [33]. This would make it possible for universities to expand around the world while also achieving higher revenues. In addition, with students' regular use of university-managed online systems, universities could obtain a greater amount of data on their behavior that would enable them to improve their decision making and strategy development.

In summary, and according to the results of the previous research, to implement BL, it is necessary to take into account several challenges, aligned with those mentioned by Khan [35] but, specifically, with what is shown in Figure 2.

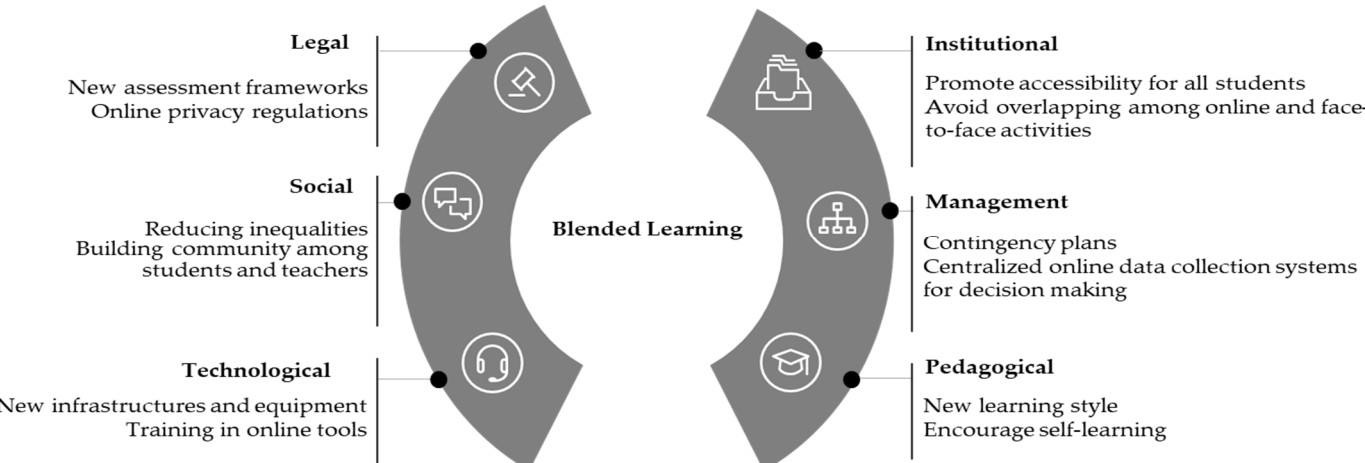

**Figure 2.** The challenges in and the course of action for implementing blended learning based on the COVID-19 experience.

Universities must facilitate the correct development of teaching and, in doing so, recognize the need to develop timetables and equipment in accordance with those teaching needs. A new and specific legal framework that guarantees privacy and ensures a suitable context for assessment is essential. Other challenges involve the teachers, who must be open to learning new tools and teaching methods based on greater communication with their students.

However, the major challenge with the most implications for the future is the ability to turn BL into a scalable teaching model. The high number of students in public universities, as well as the limited means available, leads to prioritizing those dimensions that best meet the needs of the university and the students. Therefore, a strong institutional leadership commitment to the implementation of such a model will be essential. This commitment must be long-lasting due to the complexity of the project and must seek the financial and personnel support necessary for success.

Other studies already agree with the challenges raised. The continuation of BL, or specific plans for its implementation, are necessary to be prepared for future situations, in addition to promoting the digital transformation of universities [36]. The problems of accessibility and inequality, especially in developing countries, require political intervention. However, solutions that involve the development of personalized educational tools and more effective pedagogies that do not prejudice the opportunity for learning are also being considered [37]. Other studies support the scalability of implementation, which should be accompanied by clear and precise measures to avoid overloading teachers [38], and we give special mention to one of the main problems discussed by Baloran [39] in relation to mental health during the pandemic. The issues of accessibility and socialization that BL can generate and exert on students' situations of anxiety are problems for which HEIs must have plans and tools to accompany and help students.

## 5. Conclusions

The study conducted shows BL as a teaching model that has been growing steadily to promote the development of knowledge and skills in higher education institutions [14], and that the pandemic has boosted its value. From the analysis and comparison of the results obtained from the literature, a series of conclusions about BL and its implementation can be drawn:

- The existing health situation does not seem to have altered the results obtained for BL from before the pandemic.
- The pandemic has accelerated the implementation of BL and has made it emerge as the future of universities for the globalization of teaching [40].

- Although the sudden implementation of BL has revealed the challenges that may ensue (infrastructure, lack of training, and digital divide), its positive contributions have been put into evidence (both in academic terms and resource management efficiency).
- BL can improve the quality of teaching and promote a more rational use of the technological and economic resources of universities [9] without jeopardizing academic work [41].
- BL is postulated as the first option to enrich the experience and effectiveness of student learning in new educational environments [10].
- The need to reach more students in the future while maintaining social experiences along the learning process makes BL the most viable option [16].
- The training of university faculty in the use of technology and in new teaching methodologies is the basis for the successful implementation of BL.
- Since BL is a broad concept, it is important to adopt a combination of technologies in or outside the classroom that best fits the needs of students and teachers.
- Prior research shows that the style of the teacher, the backgrounds of the students, and the subject in which the academic experience is measured can condition the results obtained [10].
- The efforts and the institutional will to implement this teaching model are necessary conditions for its correct development. BL is not about doing the same old things through a new medium [8].

In short, BL during COVID-19 developed in a complex situation, but in spite of this, BL has been able to extract the advantages of face-to-face and online teaching, demonstrating it to be a suitable teaching model and one that enhances the experience of students and teachers.

**Author Contributions:** Conceptualization, D.G. and S.B.-T.; resources, S.B.-T.; data curation, S.B.-T.; writing—original draft preparation, S.B.-T.; writing—review and editing, D.G.; visualization, D.G.; supervision, D.G.; project administration, D.G. All authors have read and agreed to the published version of the manuscript.

**Funding:** This research received no external funding.

**Institutional Review Board Statement:** Not applicable.

**Informed Consent Statement:** Not applicable.

**Acknowledgments:** This research has received the support of the Student Observatory of the Complutense University of Madrid, through Project POE 17-212.

**Conflicts of Interest:** The authors declare no conflict of interest.

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
