# Peer review of "Implementation of Blended Learning during COVID-19"

_encyclopedia, doi:10.3390/encyclopedia2040121_

Round 1

Reviewer 1 Report

I would like to commend the authors on the work they have done on this Encyclopedia article. I feel that they have some good insights into the implementation of Blended Learning during Covid-19 but could expand in some areas.

I would expand on this statement. “In universities, the implementation of BL has been prolonged more than in other levels of education as they are the least vulnerable to the need to return to the face-to-face teaching model” Why would they be least vulnerable?

I would recommend amending this statement as the survey didn’t specifically look at technology in education. “This shows that society considers the introduction of technology in education as something normal and key to education in the future, which emphasizes the importance of understanding this type of modality” Amend to “the survey showed that respondents considered blended and online teaching as normal and key” rather than the introduction of technology.

Within section 2 “description of blended learning” I feel that it would improve the paper if you were to add a paragraph about case studies of implementing blended learning during the Covid19 period. Currently the paper does not explain what actually happened during this period in terms of blended learning (rather it describes perceptions and challenges etc…), and this is the title of the entry. For example “Case studies of blended learning during Covid19 include xx xx and xx” One or two case studies of universities who did blended learning during Covid19 would improve this paper and bring more clarity on what actually happened during this period.

Figure 1 only shows some of the countries within the survey. Perhaps note this in the Figure title (i.e. Sample of responses from the survey…)

Reference 7 appears to be inaccessible? Please recheck this source.

I would define in dates the Covid19 period and the pre Covid-19 period for clarity that were analysed, in section 3.

I feel this sentence should be expanded “The results discussed above show the overlap in much of the BL research both before and during the pandemic, suggesting that the health context was not a major influence in how the students and teachers experienced BL.” Perhaps describe what this overlap is for the reader for easier readability. If possible, a table with Pre Covid Period and Covid Period with your references and key themes could be created to give greater clarity about the overlap and similarities.

I would disagree with this statement “In the case of teachers, research is scarce”. There are a good number of articles around teaching experiences during Covid 19 e.g. https://www.mdpi.com/2227-7102/11/1/19/htm I would include some of these. In addition, earlier on in the article the authors state “Moreover, it describes the consequences of BL experienced by both teachers and students during the COVID-19 pandemic.”. These are contradictory statements.

I would include “during the Covid-19 period” in this title 4.

Author Response

Thank you for taking the time to improve our manuscript. Please see the attachment.

Reviewer 2 Report

Covid-19 has forced a change in the way of teaching at every level of education. Remote learning and the use of blended learning (BL) became necessary. This situation has created a number of new dependencies affecting teaching in higher education. Some of them turned out to be useful and broadened the spectrum of teachers' workshop, others caused some additional problems most often related to the broadly understood communication. The article tries to present this topic.

The introduction describes the research background in relation to global regulations. The sources presented are a bit too meager. Since the chapter shows statistics relating to specific countries, it would be worth writing a little more about the rules related to BL there. It would also be advisable to more specifically emphasize the theses of the article and the purpose of the research.

As I understand it, the results are presented in Chapter 5. It is about challenges and opportunities of BL implementation in higher education. The chapter would need to be extended with a slightly more specific description of the dependencies mentioned. It would also be advisable to look at the risks associated with BL, which the period of the Covid -19 pandemic has highlighted very strongly. Especially those related to mental health and learning opportunities for students. A discussion would also be advisable in the article, especially since a lot of publications on similar topics have recently appeared. There are examples:

Zhang, C.; Wen, M.; Tong, K.; Chen, Z.; Wen, Q.; Yang, T.; Liu, Q. Institutional Adoption and Implementation of Blended Learning in the Era of Intelligent Education. Appl. Sci. 2022, 12, 8846. https://doi.org/10.3390/app12178846

Pokhrel, S.; Chhetri, R. A literature review on impact of COVID-19 pandemic on teaching and learning. High. Educ. Future 2021, 8, 133–141

Ngoatle, C.; Mothiba, T.M.; Ngoepe, M.A. Navigating through COVID-19 Pandemic Period in Implementing Quality Teaching and Learning for Higher Education Programmes: A Document Analysis Study. Int. J. Environ. Res. Public Health 2022, 19, 11146. https://doi.org/10.3390/ijerph191711146

In the summary of the results or in the chapter with conclusions, some recommendations would also be made regarding the further inclusion of BL in teaching in higher education.

Author Response

(The authors gave the same response as above.)

Reviewer 3 Report

The article gives a good overview of existing BL models and the dimensions of BL and  it describes the consequences of BL experienced by both teachers and students during the COVID-19 pandemic. It provides an overview of the development in BL and describes some of the challenges and opportunities.

The article exemplifies the COVID 19 period for BL, which I find difficult. As in most cases during the period of time, BL was not possible due to several restrictions. Frequently, in early 2020 instructors transitioned on-campus courses to online courses. Many of the problems stated might result from the sudden shift during quarantine. Contrasting BL to fully online (based on the experience from this period of time) is thus difficult and unspecific (Figure 1. Teaching modality that will be used in the future for higher education).  

As the article is directed at higher education, it would be interesting to see how teachers further develop the concept of BL for HE and bring in new ideas and models for teaching in BL contexts in the 21st century. Especially, now that after the pandemic, students are less prepared to come to university and higher education needs to find a good balance between online and in-person learning (Sydney, 2022) as they are now often confronted with empty lecture halls.

It would be interesting to see the concept of BL contextualized to specific learning outcomes, subject, target groups etc., in higher education, or an outline of how BL can tackle one or more of the current problems in HE, e.g., infrastructure, lack of training, digital divide  would be a major contribution.

Author Response

(The authors gave the same response as above.)

Round 2

Reviewer 1 Report

Thank you for responding to my comments, I think the paper reads much better. I just have one query based on your response "“In higher education, students do not depend on the physical presence of a teacher for their learning”". I feel that if you do include this, you would need a reference about the dependence of physical presence of teaching staff in other levels of education. 

Author Response

(The authors gave the same response as above.)

Reviewer 2 Report

The authors extended the introduction with additional explanations of the legislation in the countries to which the main part of the study relates, which is a positive change. Tables have been added and the resulting part has been significantly expanded, which made the text now more understandable and raises less doubts. I am still somewhat lacking a broader description of global BL-related regulations and research workflow, but the article may be published in its current form.

Author Response

(The authors gave the same response as above.)

Reviewer 3 Report

Thank you for sending the revised version.

Some aspects are still striking and could be addressed:

The purpose of the paper is unclear. It is stated that "The objective of this study is to show the impact of BL implementation in higher education during the pandemic." Why would that be of interest? 

There are some very strong statements, which I do not agree with and it is unclear where they come from, e.g. "In universities, the implementation of BL has been prolonged more than in other levels of education as they are the least vulnerable to the need to return to the face-to-face teaching model." or " In higher education, students do not depend on the physical presence of a teacher for their learning." This is not true especially for underprepared students or first-generation students, for example.

The additional value of Table 1. Summary of main results for each period analyzed is unclear.

Author Response

(The authors gave the same response as above.)

Round 3

Reviewer 1 Report

Thank you and well done

Author Response

Thank you very much for your comments. We are glad to know that you find the article suitable for publication.

Reviewer 3 Report

-

Author Response

(The authors gave the same response as above.)
